

# Development and validation of a multi-parameter nomogram for venous thromboembolism in gastric cancer patients: a retrospective analysis

Hang Zhou[1,2,*], Haike Lei[3,*], Huai Zhao[1,2], Kaifeng Huang[1,2], Yundong Wang[1,2], Ruixia Hong[1,2], Jishun Huo[1,2], Li Luo[1,2] and Fang Li[1,2]

[1] Department of Ultrasound in Medicine, Chongqing University Cancer Hospital, Chongqing, China
[2] Chongqing Key Laboratory for Intelligent Oncology in Breast Cancer (iCQBC), Chongqing University Cancer Hospital, Chongqing, China
[3] Chongqing Cancer Multi-omics Big Data Application Engineering Research Center, Chongqing University Cancer Hospital, Chongqing, China
* These authors contributed equally to this work.

Corresponding authors
Li Luo, 3404177743@qq.com
Fang Li, lifang0703@cqu.edu.cn

## ABSTRACT

**Objective:** Gastric cancer (GC), one of the highest venous thromboembolism (VTE) incidence rates in cancer, contributes to considerable morbidity, mortality, and, prominently, extra cost. However, up to now, there is not a high-quality VTE model to steadily predict the risk for VTE in China. Consequently, setting up a prediction model to predict the VTE risk is imperative.

**Methods:** Data from 3,092 patients from December 15, 2017, to December 31, 2022, were retrospectively analyzed. Multiple logistic regression analysis was performed to assess risk factors for GC, and a nomogram was constructed based on screened risk factors. A receiver operating curve (ROC) and calibration plot was created to evaluate the accuracy of the nomogram.

**Results:** The risk factors of suffering from VTE were older age (OR = 1.02, 95% CI [1.00–1.04]), Karnofsky Performance Status (KPS) ≥ 70 (OR = 0.45, 95% CI [0.25–0.83]), Blood transfusion (OR = 2.37, 95% CI [1.47–3.84]), advanced clinical stage (OR = 3.98, 95% CI [1.59–9.99]), central venous catheterization (CVC) (OR = 4.27, 95% CI [2.03–8.99]), operation (OR = 2.72, 95% CI [1.55–4.77]), fibrinogen degradation product (FDP) >5 μg/mL (OR = 1.92, 95% CI [1.13–3.25]), and D-dimer > 0.5 mg/L (OR = 2.50, 95% CI [1.19–5.28]). The area under the ROC curve (AUC) was 0.82 in the training set and 0.85 in the validation set.

**Conclusion:** Our prediction model can accurately predict the risk of the appearance of VTE in gastric cancer patients and can be used as a robust and efficient tool for evaluating the possibility of VTE.

## INTRODUCTION

Venous thromboembolism (VTE), including deep venous thrombosis (DVT) and pulmonary thromboembolism (PTE), is the second most common cause of death in cancerous patients and contributes to prominently extra cost (*Donnellan & Khorana, 2017*; *Lyman et al., 2021*). Meanwhile, the risk of VTE in cancer patients has not decreased despite the elevated therapeutic effect. On the contrary, some studies manifest that the incidence of VTE in cancer-associated patients has increased over the past two decades, and approximately 15% of patients with cancer will undergo VTE (*Lyman et al., 2018*; *Eichinger, 2016*).

The risk of VTE is primarily dependent on the type of cancer. Compared with other solid neoplasms, the highest incidence rate was observed in the pancreas, primary brain tumors, and gastric cancers (GC) (*Moik, Ay & Pabinger, 2020*). Apart from high incidence, the occurrence of VTE is a prognostic factor in people with cancer. Research including 3,095 advanced GC patients indicated that VTE is a remarkably predictable marker with hazard ratios (HRs) of 1.23 (95% CI [1.0–1.52]) (*Kang et al., 2012*). Given the severe consequences of VTE in cancer, several risk prediction scores have been developed to predict the likelihood of VTE occurrence. One of the most commonly used scales is the Khorana risk score (*van Es et al., 2017*). However, it should be noted that the item BMI ≥ 35 kg/m$^2$ may not be suitable for Chinese individuals, as many Chinese individuals have a relatively lower BMI, especially in gastric cancer (GC) patients. Additionally, the Khorana risk score does not take into account the impact of surgery, anticancer therapies, and supportive care treatments on VTE risk. These factors can also significantly increase the risk of VTE in cancer patients and should be considered when assessing the overall risk (*Gervaso, Dave & Khorana, 2021*).

Considering the aforementioned limitations, the objective of this study is to establish and validate a prediction model that comprehensively and predicts the likelihood of VTE in patients with gastric cancer (GC).

## MATERIALS AND METHODS

### Study population

Data were retrospectively retrieved from the database of Chongqing University Cancer Hospital in China, from December 15, 2017, to December 31, 2022. A total of 3,348 GC patients, including 1,290 newly diagnosed patients, were enrolled. The risk factors were selected based on variables previously described in the guidelines, published literature, and clinical experience. The variables below were taken into account: Patient-associated factors: sex, age, Body Mass Index (BMI), Karnofsky Performance Status (KPS), angiocardiopathy (cardiovascular disease); Cancer-associated factors: clinical stage, lymph vessel invasion; Treatment-associated factors: chemotherapy (capecitabine, oxaliplatin, CDDP, S-1 et al.), central venous catheterization (CVC) *via* jugular vein, subclavian vein, and femoral vein, operation, blood transfusion (any historical transfusion); Biomarker (gained from the latest hospitalisation): leukocyte count, platelet count, hemoglobin, albumin concentration, serum creatinine,
fibrinogen degradation product (FDP), D-dimer. Fresh blood samples were collected by drawing blood from the antecubital vein and stored in EDTA-K2-coated tubes. All blood samples were analyzed in the lab of Chongqing University Cancer Hospital. BMI, KPS, leukocyte count, platelet count, hemoglobin, albumin concentration, serum creatinine, FDP, and D-dimer were transformed into categorical variables using a specific cut-off point based on clinical experience. The informed consents were obtained from all participants and ethical approval was secured from by the ethics committee of Chongqing University Cancer Hospital (Ref: No. CZLS2023284-A).

## VTE diagnosis

According to the prevention and treatment of tumor-associated venous thromboembolism (2019 Edition), VTE includes DVT and PE. Vascular pressure Doppler ultrasound or venography was used to diagnose DVT. PE (pulmonary embolism) was diagnosed by CT pulmonary angiography (CTPA) or nuclide lung ventilation/perfusion imaging. There were no false positive results in imaging diagnosis.

## Inclusion and exclusion criteria

The inclusion criteria were as follows: (1) Patients age ≥18 years; (2) Presence of at least one hospitalization record; (3) pathology-confirmed GC including primary diagnosis or disease progression after complete or partial remission. The exclusion criteria were as follows: (1) Arterial or venous thrombosis within 3 months before admission (the latest hopitalization), and ongoing treatment with anticoagulants; (2) recent radiotherapy within the last 2 weeks, as well as chemotherapy within the past 3 months; (3) died within 2 days after admission; (4) serious data missing or incomplete. The enrolled patients were allowed to receive physical or chemical preventive anticoagulation during their hospitalization. After applying the inclusion and exclusion criteria, a total of 3,092 patients were included for further analysis. The flow diagram is shown in Fig. 1. All researchers reviewed and standardized the database in the case of unawareness of the prediction variable and outcome. The study complied with the Declaration of Helsinki and was authorized by the ethics committee of Chongqing University Cancer Hospital.

## Statistical analysis

Missing values were imputed using the 'Mice' package through multiple imputations. In order to evaluate the model objectively, the "Create Data Partition" function in the "caret" was utilized to randomly split the data set into a training set and validation set at a ratio of 3:1. The baseline characteristics between the training set and validation set were compared by the Pearson Chi-square test. Considering the large sample size and relatively small variables, we needed more variables being included in the multivariate logistic regression, we set variables with $p < 0.2$ in univariate regression analysis furtherly analyzed by multiple logistic regression as in Kilic et al. (2012). Nomogram was built to display the prediction variables and the quantitative risk assessment of VTE visually. Simultaneously, the receiver operation characteristic (ROC) curve was adopted to assess the performance of the predictive model in the training and validation set. A calibration

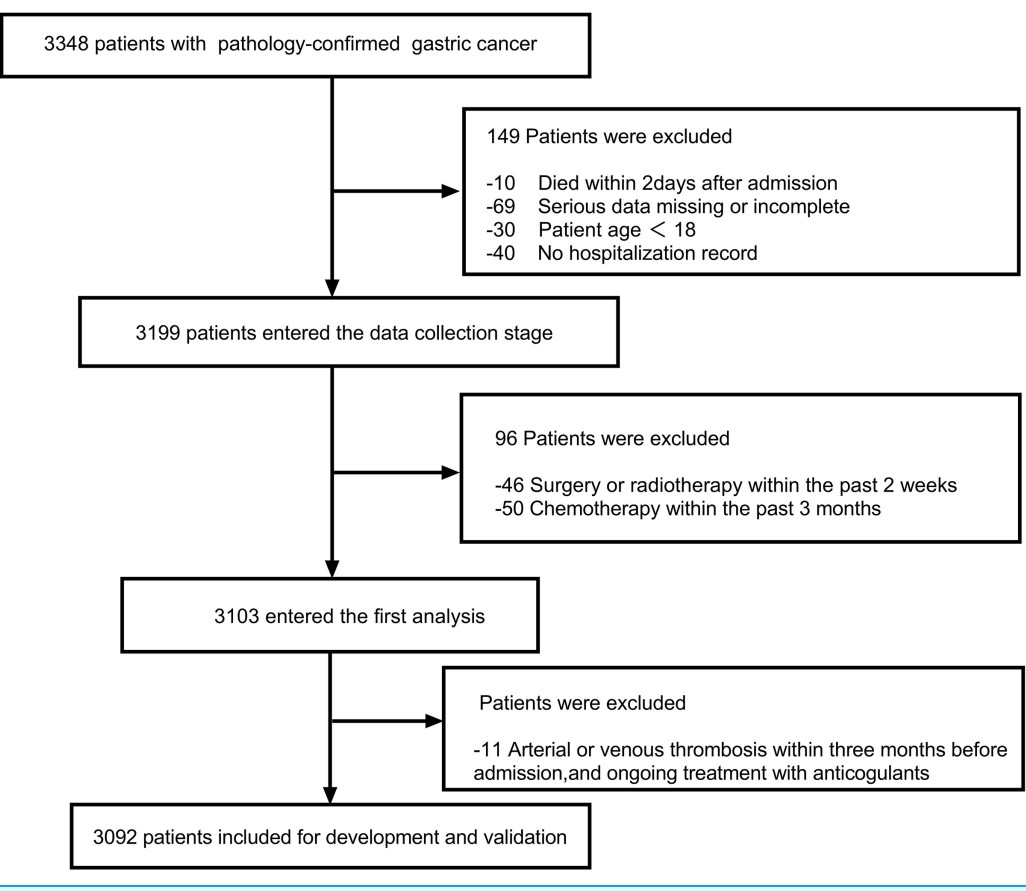

**Figure 1 The flow diagram outlining the search progress.**

plot was drawn to validate the calibration degree of the predictive model built.

All statistical analyses were conducted by R software version 4.1.0 (*R Core Team, 2022*) and SPSS software version 26.0 (IBM Corp, Armonk, NY, USA). All statistical tests were two-sided, $p < 0.05$ was statistically significant. Figures were drawn by GraphPad Prism 9 (GraphPad Software Inc, La Jolla, CA, USA).

# RESULTS

## Baseline characteristics

In the end, 3,092 gastric cancer patients were included in the study and randomly divided into a training set ($n = 2,319$) and validation set ($n = 773$) at a ratio of 3:1. There were 2,224 (71.9%) male patients and 868 (28.1%) female patients, and the median age was 62.35 ± 11.14 years old. There were 105 patients developed VTE patients, including 6 PE and 99 DVT, and the occurrence rate of which was 0.19% and 3.20%, respectively. The onset time of all these VTE ranged from 1 to 21 months, low-molecular-weight heparins (LMWHs) for the initial (first 10 days) treatment and maintenance treatment of cancer-associated thrombosis. ll parameters, including demographics, clinicopathologic features, treatment methods, and laboratory indicators, were shown in Table 1, and there were no appreciably meaningful differences between the training and validation cohorts ($p > 0.05$).

**Table 1 Comparison of factor characteristics between training set and validation set.**

| Variables | Overall 3,092 | Training set 2,319 | Validation set 773 | p-value |
|---|---|---|---|---|
| Sex | | | | |
| Female | 868 (28.07) | 648 (27.94) | 220 (28.46) | 0.817 |
| Male | 2,224 (71.93) | 1,671 (72.06) | 553 (71.54) | |
| Age (years) | 62.35 ± 11.143 | 62.268 ± 11.139 | 62.600 ± 11.159 | 0.473 |
| BMI | | | | |
| <18.5 | 590 (19.08) | 446 (19.23) | 144 (18.63) | 0.442 |
| 18.5–23.9 | 1,843 (59.61) | 1,368 (58.99) | 475 (61.45) | |
| ≥24 | 659 (21.31) | 505 (21.78) | 154 (19.92) | |
| KPS | | | | |
| <70 | 246 (7.96) | 180 (7.76) | 66 (8.54) | 0.539 |
| ≥70 | 2,846 (92.04) | 2,139 (92.24) | 707 (91.46) | |
| Angiocardiopathy | | | | |
| NO | 2,509 (81.14) | 1,876 (80.90) | 633 (81.89) | 0.577 |
| YES | 583 (18.86) | 443 (19.10) | 140 (18.11) | |
| Blood transfusion | | | | |
| NO | 2,138 (69.15) | 1,586 (68.39) | 552 (71.41) | 0.126 |
| YES | 954 (30.85) | 733 (31.61) | 221 (28.59) | |
| Clinical stage | | | | |
| I–II | 700 (22.64) | 522 (22.51) | 178 (23.03) | 0.535 |
| III | 980 (31.69) | 725 (31.26) | 255 (32.99) | |
| IV | 1,412 (45.67) | 1,072 (46.23) | 340 (43.98) | |
| Lymph vessel invasion | | | | |
| NO | 2,785 (90.07) | 2,085 (89.91) | 700 (90.56) | 0.652 |
| YES | 307 (9.93) | 234 (10.09) | 73 (9.44) | |
| Chemotherapy | | | | |
| NO | 2,054 (66.43) | 1,537 (66.28) | 517 (66.88) | 0.792 |
| YES | 1038 (33.57) | 782 (33.72) | 256 (33.12) | |
| CVC | | | | |
| NO | 2,956 (95.60) | 2,208 (95.21) | 748 (96.77) | 0.085 |
| YES | 136 (4.40) | 111 (4.79) | 25 (3.23) | |
| Operation | | | | |
| NO | 1,206 (39.00) | 887 (38.25) | 319 (41.27) | 0.148 |
| YES | 1,886 (61.00) | 1,432 (61.75) | 454 (58.73) | |
| Leukocyte count (10 ^ 9/L) | | | | |
| <11 | 181 (5.85) | 139 (5.99) | 42 (5.43) | 0.627 |
| ≥11 | 2,911 (94.15) | 2,180 (94.01) | 731 (94.57) | |
| Platelet count (10^9/L) | | | | |
| <350 | 2,765 (89.42) | 2,067 (89.13) | 698 (90.30) | 0.399 |
| ≥350 | 327 (10.58) | 252 (10.87) | 75 (9.70) | |
| Hemoglobin (g/L) | | | | |
| <100 | 868 (28.07) | 637 (27.47) | 231 (29.88) | 0.212 |

(Continued)

| Table 1 (continued) | | | | |
|---|---|---|---|---|
| Variables | Overall 3,092 | Training set 2,319 | Validation set 773 | *p*-value |
| ≥100 | 2,224 (71.93) | 1,682 (72.53) | 542 (70.12) | |
| Albumin.concentration (g/L) | | | | |
| <30 | 224 (7.24) | 160 (6.90) | 64 (8.28) | 0.229 |
| ≥30 | 2,868 (92.76) | 2,159 (93.10) | 709 (91.72) | |
| Serum creatinine (µmol/L) | | | | |
| ≤133 | 3,033 (98.09) | 2,269 (97.84) | 764 (98.84) | 0.111 |
| >133 | 59 (1.91) | 50 (2.16) | 9 (1.16) | |
| FDP (mg/L) | | | | |
| ≤5 | 2,297 (74.29) | 1,715 (73.95) | 582 (75.29) | 0.491 |
| >5 | 795 (25.71) | 604 (26.05) | 191 (24.71) | |
| D.dimer (mg/L) | | | | |
| ≤0.5 | 1,286 (41.59) | 960 (41.40) | 326 (42.17) | 0.736 |
| >0.5 | 1,806 (58.41) | 1,359 (58.60) | 447 (57.83) | |

**Note:**

Pearson Chi-square test was conducted to compare the baseline characteristics between the training set and validation set. *p* value < 0.05 was statistically significant. BMI, Body Mass Index; KPS, Karnofsky Performance Status; CVC, central venous catheterization; FDP, fibrinogen degradation product.

### The risk factors of gastric cancer

In order to find the risk factors contributing to the occurrence of VTE in GC, univariable logistic regression was first adopted to find some significant indexes in the training data set. The comparison results were considered significant when $p < 0.2$, and the corresponding indexes were included for further investigation. Finally, these elements, including age, BMI, KPS, angiocardiopathy, blood transfusion, clinical stage, CVC, operation, leukocyte count, hemoglobin, albumin concentration, serum creatinine, FDP, and D-dimer, all have significance. Simultaneously, the multivariate logistic regression analysis was used to select the most valuable elements. In the end, age, KPS, blood transfusion, clinical stage, CVC, operation, FDP, and D-dimer were identified as independent VTE risk elements and were included in the prediction model. The results of eight risk factors acquired in the multivariate logistic regression analysis were demonstrated in Table 2. The eight impact factors were shown as follows: age (OR = 1.02, 95% CI [1.00–1.04]), KPS ≥ 70 (OR = 0.45, 95% CI [0.25–0.83]), blood transfusion (OR = 2.37, 95% CI [1.47–3.84]), Clinical stage (OR = 3.98, 95% CI [1.59–9.99]), central venous catheterization (OR = 4.27, 95% CI [2.03–8.99]), operation (OR = 2.72, 95% CI [1.55–4.77]), fibrinogen degradation product >5 µg/mL (OR = 1.92, 95% CI [1.13–3.25]), and D-dimer >0.5 mg/L (OR = 2.50, 95% CI [1.19–5.28]).

### Nomogram construction for gastric cancer

The eight risk factors derived from multivariate logistic regression analysis of the training set were used to construct a nomogram to predict the risk of suffering from VTE in GC. The nomogram and online tool (https://cqcgcp.shinyapps.io/DynNomapp/) for predicting VTE are exhibited in Fig. 2. Each risk factor was endowed with scores accordingly.

**Table 2 Comparison of VTE and non-VTE in training set of gastric cancer.**

| Dependent: VTE | | NON-VTE (N = 2,237) | VTE (N = 82) | OR (univariable) | OR (multivariable) |
|---|---|---|---|---|---|
| Sex | Female | 622 (27.8%) | 26 (31.7%) | | |
| | Male | 1,615 (72.2%) | 56 (68.3%) | 0.83 (0.52–1.33, $p^* = 0.440$) | |
| Age | Mean ± SD | 62.2 ± 11.2 | 64.9 ± 10.6 | 1.02 (1.00–1.04, $p^* = 0.029$) | 1.02 (1.00–1.04, $p = 0.041$) |
| BMI | <18.5 | 423 (18.9%) | 23 (28%) | | |
| | 18.5–23.9 | 1,323 (59.1%) | 45 (54.9%) | 0.63 (0.37–1.05, $p^* = 0.074$) | |
| | ≥24 | 491 (21.9%) | 14 (17.1%) | 0.52 (0.27–1.03, $p^* = 0.062$) | |
| KPS | <70 | 160 (7.2%) | 20 (24.4%) | | |
| | ≥70 | 2,077 (92.8%) | 62 (75.6%) | 0.24 (0.14–0.41, $p^* < 0.001$) | 0.45 (0.25–0.83, $p = 0.010$) |
| Angiocardiopathy | NO | 1,815 (81.1%) | 61 (74.4%) | | |
| | YES | 422 (18.9%) | 21 (25.6%) | 1.48 (0.89–2.46, $p^* = 0.129$) | |
| Blood transfusion | NO | 1,543 (69%) | 43 (52.4%) | | |
| | YES | 694 (31%) | 39 (47.6%) | 2.02 (1.30–3.14, $p^* = 0.002$) | 2.37 (1.47–3.84, $p < 0.001$) |
| Clinical stage | I–II | 516 (23.1%) | 6 (7.3%) | | |
| | III | 713 (31.9%) | 12 (14.6%) | 1.45 (0.54–3.88, $p^* = 0.463$) | 1.16 (0.42–3.17, $p = 0.776$) |
| | IV | 1,008 (45.1%) | 64 (78%) | 5.46 (2.35–12.69, $p^* < 0.001$) | 3.98 (1.59–9.99, $p = 0.003$) |
| Lymph vessel invasion | NO | 2,008 (89.8%) | 77 (93.9%) | | |
| | YES | 229 (10.2%) | 5 (6.1%) | 0.57 (0.23–1.42, $p^* = 0.228$) | |
| Chemotherapy | NO | 1,481 (66.2%) | 56 (68.3%) | | |
| | YES | 756 (33.8%) | 26 (31.7%) | 0.91 (0.57–1.46, $p^* = 0.695$) | |
| CVC | NO | 2,137 (95.5%) | 71 (86.6%) | | |
| | YES | 100 (4.5%) | 11 (13.4%) | 3.31 (1.70–6.44, $p^* < 0.001$) | 4.27 (2.03–8.99, $p < 0.001$) |
| Operation | NO | 867 (38.8%) | 20 (24.4%) | | |
| | YES | 1,370 (61.2%) | 62 (75.6%) | 1.96 (1.18–3.27, $p^* = 0.010$) | 2.72 (1.55–4.77, $p < 0.001$) |
| Leukocyte count | <11 | 127 (5.7%) | 12 (14.6%) | | |
| | ≥11 | 2,110 (94.3%) | 70 (85.4%) | 0.35 (0.19–0.66, $p^* = 0.001$) | |
| Platelet count | <350 | 1,992 (89%) | 75 (91.5%) | | |
| | ≥350 | 245 (11%) | 7 (8.5%) | 0.76 (0.35–1.67, $p^* = 0.491$) | |
| Hemoglobin | <100 | 597 (26.7%) | 40 (48.8%) | | |
| | ≥100 | 1,640 (73.3%) | 42 (51.2%) | 0.38 (0.25–0.60, $p^* < 0.001$) | |
| Albumin concentration | <30 | 145 (6.5%) | 15 (18.3%) | | |
| | ≥30 | 2,092 (93.5%) | 67 (81.7%) | 0.31 (0.17–0.56, $p^* < 0.001$) | |
| Serum creatinine | ≤133 | 2,193 (98%) | 76 (92.7%) | | |
| | >133 | 44 (2%) | 6 (7.3%) | 3.93 (1.63–9.52, $p^* = 0.002$) | |
| FDP | ≤5 | 1,679 (75.1%) | 36 (43.9%) | | |
| | >5 | 558 (24.9%) | 46 (56.1%) | 3.84 (2.46–6.01, $p^* < 0.001$) | 1.92 (1.13–3.25, $p = 0.015$) |
| D-dimer | ≤0.5 | 949 (42.4%) | 11 (13.4%) | | |
| | >0.5 | 1,288 (57.6%) | 71 (86.6%) | 4.76 (2.51–9.02, $p^* < 0.001$) | 2.50 (1.19–5.28, $p = 0.016$) |

**Note:**
Univariate and multivariate logistic regression was conducted to determine the independent factors, in univariate logistic regression, $p^* < 0.2$ was set as cut off value for multivariate logistc regression, in multivariate logistc regression, $p < 0.05$ was set as cutoff value for independent factors. BMI, Body Mass Index; KPS, Karnofsky Performance Status; CVC, central venous catheterization; FDP, fibrinogen degradation product.

The length of the line represents the possibility of the VTE. The longer the line, the more possibility of VTE occurring. For instance, In binary variables, CVC was the most extended factor. Thus, it can be explained that the influence is the maximum. The corresponding score can transform every variable, and the summation of the scores can significantly predict the possibility of VTE.

### Validation of nomogram for VTE

The area under the curve (AUC) was drawn respectively in training data and validation data, and the AUC of the training set was 0.82 and 0.85 in the validation set, manifesting a sound discriminate capability (Fig. 3). The calibration plot was drawn to confirm if the data from Chongqing University Cancer Hospital could be well applied to the nomogram and the coefficient of determination $p$ values were 0.863 and 0.679 separately (Fig. 4). Representing that the prediction model exhibited good stability.

## DISCUSSION

A clinical prediction model based on eight variables was established for the occurrence of VTE in GC. All eight variables, including age, KPS, blood transfusion, clinical stage, CVC, operation, FDP, and D-dimer, were included in this prediction model, which can be easily collected in clinical practice. This model's discrimination, calibration, and prediction value are outstanding and have good application value in clinical recognition and decision-making. To our knowledge, there is no prediction model specially designed for GC. Consequently, it is urgent to set one. In order to formulate the most practical model, we included three aspects as patient-associated factors, cancer-associated factors, and laboratory biomarkers which impact the formation of VTE, and our importance ranking is also reasonable.

Even though CVC has a revolutionary influence on patients demanding long-term venous access will lead to a variety of components such as platelet, plasma proteins, fibrinogen, adhesion, and coagulation in the blood vessel and result in DVT (*Citla Sridhar, Abou-Ismail & Ahuja, 2020*). *Tanizawa et al. (2017)*, based on 1,140 gastric cancer patients, showed that patients who use central venous have a higher VTE incidence rate. Our result is consistent with the study mentioned above.

Evidence from extensive cohort studies indicates that the cancer stage is a risk factor for VTE (*Gervaso, Dave & Khorana, 2021*). Data on 671 gastric cancer patients found that disease stages were significantly associated with VTE (OR = 2.24, 95% CI [1.42–3.53]) (*Abdel-Razeq et al., 2020*). A retrospective study including 2,085 gastric cancer patients found a prominently different VTE incidence rate in different stages (*Lee et al., 2010*). However, *Fuentes et al. (2018a)* did not find any relationship between VTE and stage. Similarly, *Wada et al. (2017)* includes 976 gastric cancer patients divided the stage into I and no less than II stage, and there is no difference between the two cohorts. Our research found that VTE ratios were significantly higher in the IV stage than in the I–II stage. The reason is that many advanced cancer patients' conditions worsen with less activity and more combination therapy, which all contribute to VTE (*Streiff et al., 2021*). As for no meaning in the two researches above, the possible cause may be a relatively small sample

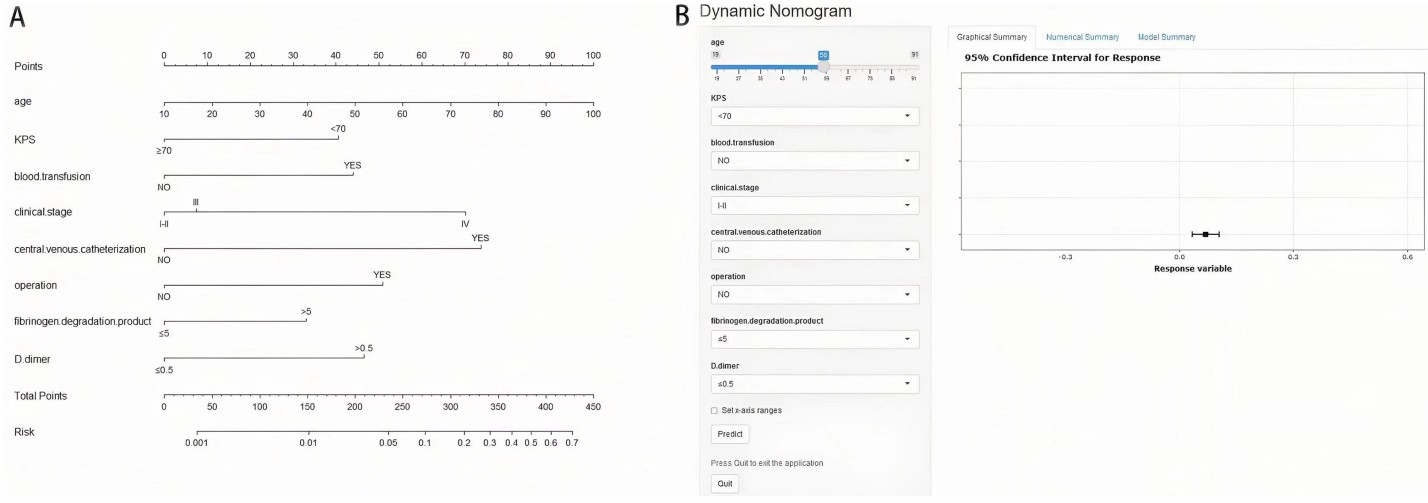

**Figure 2 Nomogram (A) and online tool (B) of prediction model in gastric cancer.**

size and a significant quantity difference between stages I and IV (64% *VS* 3%), which may confuse the reality.

As one of the most essential and standardized therapeutic choices in GC, the operation has played a well-known contribution to VTE occurrence (*Gervaso, Dave & Khorana, 2021*). Surgical interventions can not only damage endothelial cells, which can activate the coagulation system but also stimulates sympathetic nerve by pain and increase the time spent lying on the bedbound (*Khan et al., 2021*). Previous studies about cancer patients indicated that the VTE rate in the overall 30 days after major surgeries ranges from 1.8–13.2%, highlighting the effect of surgery (*Hammond et al., 2011*). The results of our study support this idea.

There are various research viewpoints on blood transfusion. A meta-analysis including more than three million participants (*Wang et al., 2021*) found that preoperative blood transfusion is a risk factor for postoperative VTE (OR = 2.95, 95% CI [1.63–5.30]). Nonetheless, *Baumann Kreuziger et al. (2021)* found that there is no association between red blood cell (RBC) transfusion with increased VTE risk in 657,412 hospitalized patients (OR = 1.0, 95% CI [0.96–1.05]). In this retrospective study, blood transfusion contributes to enhanced VTE risk (OR = 2.37, 95% CI [1.47–3.84]). Up to now, there is not a definite cause to explain this phenomenon. One reason is that RBCs undergo structural and biochemical changes and hemolysis at 4 °C, which accelerate thrombosis (*Baumann Kreuziger et al., 2021*). Another reason is that free hemoglobin may further accelerate VTE because it can blind nitric oxide, which plays an essential role in vasodilator and inhibitor of gathering adhesion (*Kanias et al., 2017*). Meanwhile, a study found that thrombosis was dose-dependent (*Xenos, Vargas & Davenport, 2012*). Therefore, taking transfusion into account and not considering other elements such as platelet, hemoglobin, and dosage in blood transfusion is not comprehensive.

D-dimer, a degradation product of cross-linked fibrin, increases quickly in acute thrombosis. However, it can also rise in conditions such as cancer, infection, and

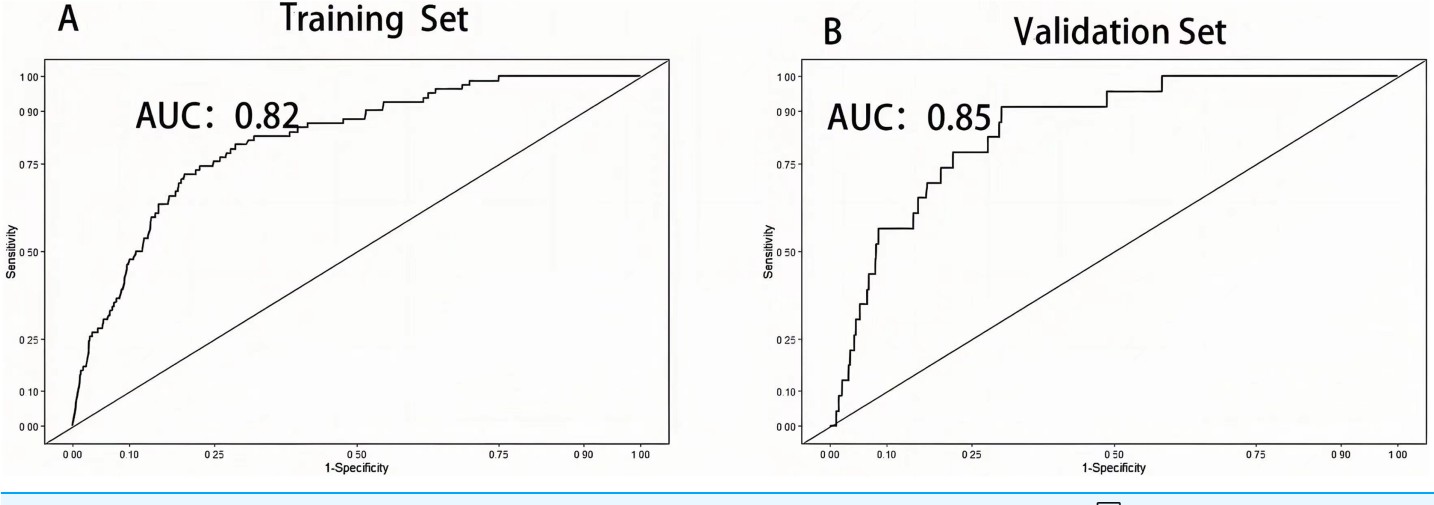

**Figure 3 Receiver operation characteristic curve of training set and validation set.**

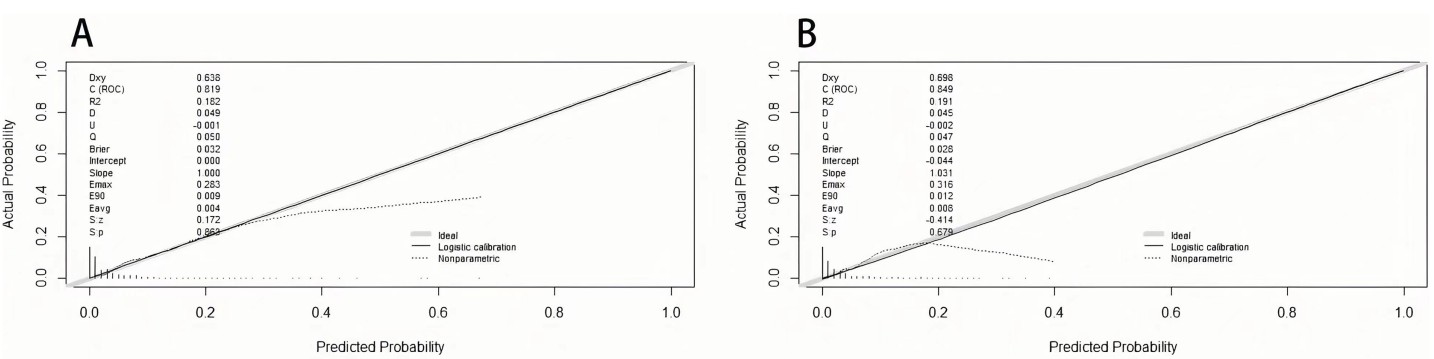

**Figure 4 The calibration plot of training set (A) and validation set (B).**

inflammation (*Khan et al., 2021*). A previous study in cancer found that high D-dimer levels were connected with increased thrombus risk (*Khorana et al., 2021*). However, a study including 110 patients with gastrointestinal cancer found no significant D-dimer level difference between patients with and without VTE (*Kimura et al., 2016*). Our data indicated that D-dimer could be used as a prediction index for gastric cancer. Small samples and different cancer types may casuse confusion.

FDP is the production after fibrinogen degradation and is an indicator reflecting the hyperactivity of the fibrinolytic system. When coagulation is activated, fibrin monomers will be changed into insoluble fibrin networks (*Lord, 2011*). Elevated plasma fibrinogen level dramatically increases the risk of VTE (*Hylckama Vlieg & Rosendaal, 2003*). To the best of our knowledge, this study first used FDP to predict VTE risk in gastric cancer and found it to be an ideal index.

Studies found that increasing age is a risk factor for VTE in cancerous people (*Streiff et al., 2021*). A previous study indicated that patients over 70 years old suffering from chemotherapy had a 2-fold risk of thrombus than younger patients (*Vergati et al., 2013*). On the contrary, *Abdel-Razeq et al. (2020)* found that age has no impact on the VTE rate.

However, the result deserves consideration because his article only divided age into two groups according to if older than 50 years old. As is known to all, age contributes to cancer and worsens health conditions, contributing to VTE (*Gervaso, Dave & Khorana, 2021*). Our results also verify this idea. In regard to the research of *Abdel-Razeq et al. (2020)*, we do not think the year of 50 is a reasonable boundary because there isn't a significant difference between people over and below 50 years old.

KPS has been widely used in medical oncology as a simple and valuable method to evaluate daily performance in an activity. Less activity means more time to lie in bed and more opportunities to suffer VTE. Our result also supports this idea.

Some studies found that the VTE occurrence rate is 9–12% (*Fuentes et al., 2018b*). However, other studies reported that a 1-year incidence of VTE was 3.5% in gastric cancer under chemotherapy (*Kang et al., 2012*). In this retrospective study, The rate of thrombus is 3.4%, lower than a majority of studies. The main reason is that our hospital pays more attention to VTE prevention, and not only operation patients but also comprehensive therapy patients will all be tested by thrombus scales and given corresponding therapy. Otherwise, the follow-up period also plays an important role in the VTE rate, which should not be neglected. In this study, we pay more attention to the VTE rate in hospitalization.

Another reason is racial variations which also plays a non-negligible effect. For example, Asian-pacific patients have a dramatically lower risk of developing VTE than Caucasians (*Chew et al., 2006*). Some indicators, such as BMI, leukocyte count, platelet count, hemoglobin, and albumin concentration included in Khorana, didn't show a significant difference in our study. There are some reasonable reasons for the following. Weight loss is found in up to 62% of gastric cancer (*Wanebo et al., 1993*). Therefore, it is unreasonable to use it as a prediction item. Leukocyte count and platelet rise are also not common in many advanced cancer patients because myelosuppression is very common in patients undergoing treatment. As for hemoglobin, a previous study found that bleeding happens in less than 20% of GC patients, which does not always lead to hemoglobin levels of less than 100 g/L, limiting the usage (*Wanebo et al., 1993*). As for albumin concentration, albumin is an inflammation marker, patients with elevated inflammation often have lower albumin levels and, consequently, a heightened risk for VTE (*Chi et al., 2019*).

This article first sets up a prediction model based on possible influence factors in the whole therapeutic process to predict the risk of thrombus in gastric cancer, and the effect is good. Otherwise, it also implies that the most used Khorana score is not an excellent choice to predict VTE in GC because it was developed and validated in a mixture of solid tumors receiving chemotherapy (*Khorana et al., 2021*).

Although our prediction model is robust, There are some shortcomings in the study. Firstly, the outcome included DVT and PTE, so we cannot obtain the valid prediction model separately. Secondly, the cutoff value to separate the two groups was based on clinical experience and literature, which may not be the optimal choice. Thirdly, this is a retrospective study, some variates like the history of antiplatelet medications, smoking status, previous history of VTE was very difficult to include. Finally, the prediction model was developed and validated in a single center, the model lacks external validation, which is an area we need to improve upon in the future.

## CONCLUSION

VTE is a common and deadly syndrome during the whole therapeutic process of GC. In this study, the incidence rate is 3.4%. A novel prediction model was developed and validated in 3,092 gastric cancer patients and illustrated well prediction value for screening high-risk thrombus. However, it was conducted in retrospective analysis, and prospective validation is needed to strengthen this conclusion.

### Funding

This work was supported by the National Cancer Center climbing fund, China (No. NCC201822B75), the Chongqing Technology Innovation and application development project, China (No. cstc2019jscxmsxmX0099) and the Natural Science Foundation of Chongqing, China (No. cstc2020jcyj msxmX0538). The funders had no role in study design, data collection and analysis, decision to publish, or preparation of the manuscript.

### Grant Disclosures

The following grant information was disclosed by the authors:
National Cancer Center climbing fund, China: NCC201822B75.
Chongqing Technology Innovation and application development project, China: cstc2019jscxmsxmX0099.
Natural Science Foundation of Chongqing, China: cstc2020jcyj msxmX0538.

### Competing Interests

The authors declare that they have no competing interests.

### Author Contributions

- Hang Zhou conceived and designed the experiments, analyzed the data, authored or reviewed drafts of the article, and approved the final draft.
- Haike Lei conceived and designed the experiments, authored or reviewed drafts of the article, and approved the final draft.
- Huai Zhao performed the experiments, prepared figures and/or tables, and approved the final draft.
- Kaifeng Huang performed the experiments, analyzed the data, prepared figures and/or tables, and approved the final draft.
- Yundong Wang conceived and designed the experiments, prepared figures and/or tables, and approved the final draft.
- Ruixia Hong conceived and designed the experiments, performed the experiments, prepared figures and/or tables, and approved the final draft.
- Jishun Huo performed the experiments, prepared figures and/or tables, authored or reviewed drafts of the article, and approved the final draft.
- Li Luo conceived and designed the experiments, authored or reviewed drafts of the article, and approved the final draft.

- Fang Li analyzed the data, authored or reviewed drafts of the article, and approved the final draft.

## Human Ethics

The following information was supplied relating to ethical approvals (*i.e.*, approving body and any reference numbers):

Ethics Committee of Chongqing University Affiliated Tumor Hospital.

## Data Availability

The raw data is available in the Supplemental File.

## Supplemental Information

Supplemental information for this article can be found online at http://dx.doi.org/10.7717/peerj.17527#supplemental-information.

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
