# Peer review of "Development and validation of a multi-parameter nomogram for venous thromboembolism in gastric cancer patients: a retrospective analysis"

_PeerJ, doi:10.7717/peerj.17527_

## Round 0.1 · original submission · Major Revisions

Reviewers provided valuable insights and offered impactful feedback on this manuscript. Kindly incorporate their suggestions and carefully address all comments in your manuscript. Thank you.

**Language Note:** The review process has identified that the English language must be improved. PeerJ can provide language editing services - please contact us at [email protected] for pricing (be sure to provide your manuscript number and title). Alternatively, you should make your own arrangements to improve the language quality and provide details in your response letter. – PeerJ Staff

·

Basic reporting

Generally the manuscript is well written and sufficient background and context were provided.
I have some suggestions for improvement.
I would recommend reviewing for formatting and language.
Some extra spaces (e.g. lines 121, 161) and different font/sizes.
Line 170: check the context for using the word public.
Line 166: move the figure number to the end of the sentence.

Experimental design

The methods are well explained. I researchers set a p-value of 0.2 for variables to be included in the regression model, I would recommend including a justification of this approach and citing source.

Validity of the findings

I would suggest adding the statistical test used and statistical significance cut-off values in a footnote for each table.

Figure 2: add a caption noting how to interpret the results.

Additional comments

No comment.

Reviewer 2 ·

Basic reporting

English needs improvement.

Experimental design

No comment.

Validity of the findings

The results are meaningful.

Additional comments

In this manuscript, Zhou et al. have conducted a thorough exploration and validation of risk factors for VTE in gastric cancer patients, utilizing data from a single center. They have successfully developed a nomogram and a Shiny application for practical use. Despite extensive prior research on cancer-associated VTE, this study contributes valuable insights, particularly concerning the Chinese population. The manuscript is clear and comprehensible. However, there are several critical aspects that warrant further elaboration:

1. The authors should elaborate on the specifics of VTE. Key details such as the onset time of VTE following the diagnosis of gastric cancer, the incidence of pulmonary embolism versus deep vein thrombosis, and the approaches taken for VTE treatment need to be detailed.
2. Further information on risk factors is required. For instance, in the case of blood transfusions, it should be clarified whether the reference is to any historical transfusion or those occurring within a specific timeframe relative to the onset of VTE. Clarification is also needed regarding the timing of measurements for FDG/D-dimer and other laboratory variables, as the current presentation is ambiguous.
3. The study seems to overlook certain significant risk factors for VTE, such as the usage of antiplatelet medications (like aspirin and clopidogrel), smoking status, and any previous history of VTE (beyond a 3-month period).
4. The English language usage throughout the manuscript requires systematic improvement. For example:
Line 52: Replace "second death cause" with "second most common cause of death."
Line 60: Change "the appearance of" to "the occurrence of."
Line 82: Clarify if "angiocardiopathy" refers to cardiovascular disease.
Line 136: Use "risk factors" instead of "risk factor."
Line 145: Replace "defined" with "identified."

Minor Points:
- Line 35 (results section of the abstract): The risk factors should be specified as "older age" and "advanced clinical stage."
- Line 98: Clarify the statement, "there were no false positive results in imaging diagnosis," as V/Q scans are known for high false-positive rates.
- Line 103: Elucidate what is meant by 'admission' and address the scenario of patients with multiple admissions.
- Line 104: Rationale for excluding surgeries within the last two weeks should be revisited, considering that a VTE is still regarded as provoked if major surgery occurred up to three months prior to the VTE event.
- Line 241: Provide clarity on the reference to "Hikmat's" research.
- Line 264: Explain that albumin is also an inflammation marker. Patients with elevated inflammation often have lower albumin levels and, consequently, a heightened risk for VTE.

Reviewer 3 ·

Basic reporting

1. The authors do not provide the code of the R script. I suggest authors provide reproducible code files as supplementary so that readers can reproduce the result of the work. Correspondingly, I suggest authors list the versions of the R packages.
2. There is a typo in supplementary material: "raw date" should be raw data.
3. Please increase the resolution of Figures 2-4. I cannot see the words clearly.
4. In lines 96-97, the "PE" should be spelled out for the first occurrence.
5. In line 117, please provide the explanation for the threshold P<0.2. Usually, the significant level is set below 0.1
6. In line 161, the first letter of the sentence is not capitalized.

Experimental design

The machine learning process should be improved.
In the machine learning process, the dataset should be split into: a training set for variable selection and training, a validation set for validating the variable selection and deciding the hyper-parameters of the model like training stop criterion, and a test set to be set aside until testing. However, I do not find a testing set defined in the manuscript. maybe the authors use the validation set for testing. The performance will be overestimated in this way because the information of the test set has been leaked to the model through the feature selection and validation process.

Validity of the findings

The performance of the model has been overestimated. Please follow the workflow of the machine learning process.

Reviewer 4 ·

Basic reporting

Author built up a logitic regression model to predict the highest venous thromboembolism (VTE), in the considering of gastric cancer. Samples from 3092 patients across 5 years were used in the model training and testing. A relatively high AUC was reported and some key factors such as age, blood transfusion and central venous catheterization were detected by analysis.

Experimental design

1. In feature selection, authors applied the univariable logistic regression to each feature seperately. However, the P=0.2 threshod needs more explanation. And authors need provide more evidence for not using other feature selection method such as recursive feature elimination or Lasso.

2. In model evaluation, samples are imbalanced w.r.t several features. Besides dividing data into training and testing, authors may consider the cross-validation to reducing the evaluation noise.

Validity of the findings

1. All figures are too small, please make them large for review.

2. It's hard to measure the predicting power of this model based on one metric like AUC. Authors may consider more metrics for model evaluation and consider another benchmark model as comparison.

---

## Round 0.2 · accepted · Accept

The authors have conducted excellent work on this manuscript, incorporating feedback from reviewers to enhance the clarity and depth of the content. They have successfully elucidated the Development and validation of a multi-parameter nomogram for venous thromboembolism in gastric cancer patients. Thanks to the thoughtful integration of the reviewers' suggestions.

Reviewer 2 ·

Basic reporting

None

Experimental design

None

Validity of the findings

None

Additional comments

The authors have satisfactory addressed all my questions.

Reviewer 3 ·

Basic reporting

The issues have been improved.

Experimental design

The issues have been improved.

Validity of the findings

The issues have been improved.

Reviewer 4 ·

Basic reporting

The authors made the corresponding in the revised version, the manuscript should be accepted.

Experimental design

No comment

Validity of the findings

No comment